# Anisakid Nematodes and Potential Risk of Human Anisakiasis through the Consumption of Hake, *Merluccius* spp., Sold Fresh in Spanish Supermarkets

**DOI:** 10.3390/pathogens11060622

**Published:** 2022-05-26

**Authors:** Màrius V. Fuentes, Elena Madrid, Clara Cuesta, Carla Gimeno, Marta Baquedano-Rodríguez, Isaac Soriano-Sánchez, Ana María Bolívar, Sandra Sáez-Durán, María Trelis, Ángela L. Debenedetti

**Affiliations:** Parasites & Health Research Group, Departament de Farmàcia i Tecnologia Farmacèutica i Parasitologia, Facultat de Farmàcia, Universitat de València, Av. Vicent Andrés Estellés s/n, 46100 Burjassot, València, Spain; elena.madrid@uv.es (E.M.); claracuesta22@hotmail.com (C.C.); carlagmol95@gmail.com (C.G.); marta.baquedano.91@gmail.com (M.B.-R.); isaac.sanchez.uv@gmail.com (I.S.-S.); ambolivar@hotmail.com (A.M.B.); sandra.saez@uv.es (S.S.-D.); maria.trelis@uv.es (M.T.); aldebenedetti@gmail.com (Á.L.D.)

**Keywords:** hake, *Merluccius merluccius*, *Merluccius bilinearis*, Spain, Ascaridoidea larvae, *Anisakis*, human anisakiasis risk, preventive measures

## Abstract

Nematode parasite species belonging to the *Anisakis simplex* complex are the most important cause of human anisakiasis through the consumption of (mainly) undercooked, previously not frozen, or conveniently treated fish. In Spain, the consumption of hake has been recognized as an important source of this parasitosis. With the aim of shedding light on the risk factors that can influence the potential risk of human anisakiasis in Spain through the consumption of fresh hake sold by nationwide supermarket chains, a total of 536 small hake specimens belonging to the species *Merluccius bilinearis* caught off the Northeast American coasts and *Merluccius merluccius* caught in the Northeast Atlantic and Mediterranean waters was analysed. Anisakids morphologically identified as *Anisakis* type I were found as the most prevalent and the most abundant species and were considered the main potential cause of human anisakiasis. Intrinsic and extrinsic factors concerning the hake, such as its origin and season of capture, its size, as well as the days passed between its capture and consumption, should be taken into account to avoid this human parasitosis. It is essential that consumers have access to fish label information which should include, as regulated by the European Commission, traceability data.

## 1. Introduction

Human anisakiasis, the most frequent form of human anisakidosis, has been recognized as an emerging food-borne disease worldwide, especially in Japan and in some Western European countries such as Spain and Italy, where fish consumption is high and consumers are able to purchase fresh and uneviscerated fish (in some of which Anisakidae larvae are very common) [1,2].

Spain seems to have the highest incidence in Europe [3], with about 8000 annual cases estimated in recent years, although the anisakiasis burden in the human population is unknown due to of the scarcity of epidemiological data [4], with the traditional consumption of raw anchovies, *Engraulis encrasicolus*, in vinegar posing the main source of risk [5], although the consumption of other fish species has also been reported multiple times as the cause of human anisakiasis in Spain.

*Anisakis* spp. larvae can be found parasitizing a wide range of marine teleost species inhabiting the Atlantic and the Mediterranean, as well as the Pacific to the Antarctic area, affecting overall fish product quality [6].

So far, among the nine species of *Anisakis* genetically detected [7], *A. simplex* (*s.s.*) and *A. pegreffii* are recognised as zoonotic species causing human anisakiasis [8].

The L_3_ of the genus *Anisakis* are very similar morphologically, traditionally classified as type I or type II larvae, with the species within each type being morphologically indistinguishable [9]. They can, however, be differentiated molecularly and have been grouped into four clades [10,11].

Nematodes belonging to the family Anisakidae (such as species of the genera *Anisakis*, *Pseudoterranova* and *Contracaecum*) have a complex aquatic biological cycle, which includes microscopic crustaceans of the zooplankton as the first intermediate host, fishes and cephalopods as secondary intermediate or paratenic hosts, and cetaceans, pinnipeds and fish-eating birds as definitive hosts. Moreover, there are other nematode species belonging to the genus *Hysterothylacium* (Raphidascaridae), very close to anisakids, which frequently parasitize very commonly consumed fish [12]. Cases of allergies to these parasites have also been considered as anisakiasis [13].

Consumers can be infected through the ingestion of fish or squids, harbouring the infective L_3_ larvae no matter whether the food item is consumed raw, undercooked, salted, marinated, or previously not frozen or conveniently treated by other culinary methods.

Although the disease may appear in different forms, namely gastric, intestinal, ectopic and allergic (with urticaria/angioedema and anaphylaxis), or even occupational asthma and arthritis [14] and, as recently suggested [15], *Anisakis* could have tumorigenic potential. Anisakidosis should also be considered as an occupational disease, with fishermen and handlers being at risk of developing asthma caused by the inhalation of nematode antigens [16,17].

The high prevalence of larvae of anisakid nematodes in some commercially available fish species also has, in addition to the anisakidosis risk for consumers, economic implications, as the presence of these larvae impacts the quality of the fish and its aesthetics, with the ensuing rejection of parasitized fish by consumers, thus causing significant losses to the fishing industry [18,19].

The European hake, *Merluccius merluccius*, is an abundant species caught throughout the Northeast Atlantic and the Mediterranean which is commonly sold in European markets, thus making it an economically important fish species. According to a study carried out by FROM (Spanish Market Regulation and Management Fund), in Spain 94% of households declare that they buy hake. Due to this high demand, the number of European hake caught in the Atlantic and in Mediterranean waters is not high enough to satisfy the demand on the Spanish market, making it necessary to import other species of hake (such as the silver hake, *M. bilinearis*, among others, caught off the Northeast American coasts) which can be acquired at various Spanish supermarkets and are also fresh and uneviscerated [20].

The European hake is distributed in the Eastern Atlantic, including Norway and Iceland, southward of Mauritania, the Mediterranean Sea and along the southern coast of the Black Sea. The north-east Atlantic and the Mediterranean Sea populations of *M. merluccius* have been considered as genetically separated stocks [21]. Moreover, in the Atlantic waters of the EU there are two stocks of hake: the northern stock, and the southern stock off the Atlantic coasts of Spain and Portugal [22]. The silver hake is distributed in the Northwest Atlantic, including the coasts of Canada and USA from the Bell Isle Channel to the Bahamas; it is most common from southern Newfoundland to South Carolina. Both species of hake are demersal and oceanodromous, with a depth range between 30 and 1075 in *M. merluccius* and between 55 and 914 m in *M. bilinearis*. Adults feed mainly on fish (small hakes, anchovies, pilchards, herrings, cod fishes, sardines and gadoid species) and squids. The young feed on crustaceans (especially euphausiids and amphipods) [23].

These voracious predator fish may be infected with *Anisakis* spp. larvae through the consumption of small crustaceans or even other smaller infected fish. Although Spanish cuisine includes a great variety of ways to prepare hake, in many of them the fish is not sufficiently well cooked to inactivate anisakid larvae potentially present in its flesh, which makes it a potential source of human anisakiasis, with some human cases having been reported due to its consumption such as the three recently reported cases in Barcelona [24]. Previous studies have reported high levels of *Anisakis* larvae in the European hake, mainly from the North East Atlantic [22]. Moreover, the two most often reported *Anisakis* species in *M. merluccius* originating from the North East Atlantic and the Mediterranean are *A. simplex* (*s.s.*) and *A. pegreffii*, respectively, both belonging to the morphotype I [2,22,25,26,27]. In *M. bilinearis*, the scarce studies carried out have reported *Anisakis* type I larvae, some of them identified molecularly mainly as *A. simplex* (*s.s.*) but also, surprisingly, as the hybrid genotype *A. simplex* (*s.s.*) × *A. pegreffii* also [12,28].

Epidemiological studies carried out to determine which commercially available fish species are most parasitized and which fishing grounds have the highest rates of parasitization are of great interest [29]. Although there are many reports on the population dynamics of anisakid nematodes, mainly *Anisakis* spp. In the Mediterranean and in the Atlantic, an updated assessment of infection levels in commercially available fish species, such as *Merluccius* spp. In this case, is crucial to assess the human anisakidosis risk [30].

Consequently, the main objectives of this study are: (1) to assess the risk of human anisakidosis, mainly anisakiasis, through the consumption of *M. merluccius* and *M. bilinearis* sold by nationwide Spanish supermarket chains, by means of the analysis of the potential influence of intrinsic and extrinsic factors of these fish species on the presence and the worm burden of anisakid larvae; (2) to inform consumers on the importance of the information on fish crate labels to prevent and diminish the human anisakidosis risk; and (3) to inform the institutions dealing with the fish trade and food safety admitted to the market about the parameters of infestation of commercially sold fish from the fishing areas included in the study. Moreover, since the most common anisakid species reported in hake cannot be easily distinguished visually by consumers, fishermen, and handlers, only morphological and not molecular identification has been carried out, differentiating the larvae found at genus level and at morphotype level (types I and II) in the case of *Anisakis*. Our study hypotheses are that: geographic origin, size (weight and length) and seasons of capture may influence the prevalence and the abundance of anisakid larvae in hakes; and that the days passed since capture may influence migration of larvae from viscera to the flesh in previously parasitized fish. If these hypotheses are verified, consumers would have additional criteria at their disposal when choosing fish, considering the information on fish crate labels. The European Union makes provision of this information at the selling point obligatory, with the aim to complement prevention measures and ultimately diminish the consumer risk of parasitization.

## 2. Results

The three populations of hake were found parasitized by Ascaridoidea nematode larvae, with the Atlantic population of the European hake being the most parasitized (87.8%), followed by the silver hake (65.7%) and the Mediterranean population of the European hake (26.0%) (Table 1). However, the Mediterranean European hake presented a higher abundance than the silver hake, although the parasite load of the Atlantic European hake was much higher than that of the other two (Table 2).

Three clearly different Ascaridoidea morphotypes were found in the three hake populations, which were morphologically classified as family Anisakidae (*Anisakis* type I larvae and *Contracaecum* spp.) and family Raphidascaridae (*Hysterothylacium* spp.). *Anisakis* type I was the most prevalent and abundant morphotype in the three populations, with the two other morphotypes being much rarer in terms of prevalence and abundance (Table 1 and Table 2). Additionally, two European Mediterranean hakes were found parasitized by larvae of *Anisakis* type II, one of which carried two larvae in the viscera and the other one larva in the flesh.

Parasitization by *Anisakis* type I larvae was also higher in the Atlantic population of the European hake than in the two other populations (Table 1), with this difference being statistically significant (χ^2^ = 165.752; *p* < 0.0001). The Odds Ratio (OR) was 22.82 (12.98–40.12 95% CI) with respect to the Mediterranean population and 4.48 (2.58–7.76) with respect to the silver hake population. Moreover, the OR of the Northwest American population with respect to the Mediterranean was 5.10 (3.24–8.01).

Likewise, the mean abundance of *Anisakis* type I larvae was also higher in the Northeast Atlantic population than in the two other populations (Table 2), also with statistically significant differences: U = 6638.0; *p* < 0.0001 when compared to the silver hake population; and U = 5560.0; *p* < 0.0001 when compared to the Mediterranean population. However, surprisingly, as was appreciated in the total of Ascaridoidea larvae, the mean abundance of the Mediterranean hake population was higher than that of the silver hake population (U = 11333.5; *p* < 0.0001).

A higher presence of *Anisakis* type I larvae in the viscera than in the flesh was found in the three populations of hake analysed (Table 3), and with statistically significant differences in the Atlantic European hake (χ^2^ = 4.704; *p* = 0.044) (OR = 1.67; 1.04–2.67) and in the silver hake populations (χ^2^ = 165.752; *p* < 0.0001) (OR = 2.87; 1.83–4.515). When comparing the abundance between viscera and flesh, the highest larvae burden was found in the viscera in the Mediterranean (Z = 4.462; *p* < 0.0001) and in the silver hake (Z = 5.941; *p* < 0.0001) populations. However, in the Atlantic European hake population the mean abundance was higher in the flesh than in the viscera (Table 4), also containing a significant statistical difference (Z = 2.190; *p* = 0.029).

### Influence of Intrinsic and Extrinsic Factors

Concerning the season of capture, the prevalence and the mean abundance of *Anisakis* type I larvae were, in the three hake populations, higher in the specimens caught in autumn-winter than those caught in spring-summer (Table 3 and Table 4). However, only the differences of mean abundance of the Atlantic European hake (U = 1692.5; *p* < 0.0001) and the silver hake (U = 2705.0; *p* = 0.029) showed statistically significant differences.

Considering the two groups of weight and length in which each of the three populations of hake were divided (see Material and Methods section), *Anisakis* type I parasitization was always higher in the group of a larger host size. However, there were statistically significant differences concerning the prevalence in *M. merluccius* only, both in the Atlantic hake and in the Mediterranean hake populations. Concerning the mean abundance, the statistically significant differences were found in the three populations of hake (Table 5 and Table 6).

The higher helminth burden in the weight and length groups of the three populations of hake was confirmed when studying the influence of quantitative host size, with positive correlations related to length and weight and all of them statistically significant (Table 7).

Moreover, the proportion of *Anisakis* type I larvae in the flesh related to the fish size of the three populations of hake analysed showed statistically significant negative correlations (Table 8).

Regarding the presence of parasites in the flesh and the number of days passed between the catch and the analysis of the specimens, the results of the Binary Logistic Regression show that the prevalence of *Anisakis* type I increases with the days passed after the catch of the hake in the Atlantic hake (χ^2^ = 15.304; *p* = 0.004) and in the Mediterranean hake (χ^2^ = 23.849; *p* < 0.0001) populations. However, in the case of the silver hake population this positive relation was observed considering the spring-summer subpopulation (χ^2^ = 8.342; *p* = 0.015) only.

## 3. Discussion

### 3.1. Global Parasitization

The results obtained in this regional study from València can easily be extrapolated to the whole of Spain, as the supermarket chains surveyed are nationwide. Although they tend to sell fish from local and nearby fishing grounds, from the Atlantic or the Mediterranean, depending on their geographical location in the Iberian Peninsula, other fish stocks are usually also offered at the selling points originating from almost the same fishery areas.

The very high global prevalence of *Hysterothylacium* and *Anisakis* type I larvae in the samples of European Atlantic hake and silver hake, as well as the low prevalence in the sample of European Mediterranean hake stands out. These results are in agreement with the most of the previous studies carried out on samples of the European Atlantic hake [12,22,25,27,31,32,33,34,35,36], the silver hake [37], and the European Mediterranean hake [12,25,33,38,39,40]. In this last case, the studies carried out in the same Mediterranean subdivision, FAO 37.1.1—Balearic, were taken into account, with the exception of the study carried out by Valero et al. [26], which found a higher prevalence of parasitization in this area. However, the higher mean abundance of *Anisakis* type I found in the Mediterranean sample compared to the silver hake sample stands out.

This higher parasitization of *M. merluccius* in the fishery grounds of the Atlantic Ocean compared with Balearic division has also been reported in previous studies [12,25], as well as in other fish species too. This variation of the parasitization parameters in both geographical locations could be the result of various factors [26,41] such as the infection level of marine mammals, the definitive hosts of anisakids; the abundance of intermediate and paratenic hosts; and the role of each fish stock in the trophic chain (especially in the case of hake).

The presence of larvae belonging to the genera *Hysterothylacium* and *Contracaecum* was reported previously by other authors in some of the three hake populations analysed [12,28,37]. However, as far as we are aware, the report of *Contracaecum* in *M. merluccius*, both in European and Mediterranean hakes, as well as *Hysterothylacium* in M. bilinearis, are new records for this fish species.

In an analysis made on a smaller part of the samples analysed in the present research, the molecular analysis of some anisakid larvae was carried out [28] as previously described by Madrid et al. [1], identifying the following *Anisakis* species: *A. simplex* (*s.s.*), *A. pegreffii* and the hybrid genotype of both species, in the viscera and the flesh of the European Atlantic hake; *A. simplex* (*s.s.*) and the above-mentioned hybrid genotype in the viscera and the flesh of the silver hake, and *A. pegreffii* in the flesh of the European Mediterranean hake, although only one larva could be analysed from this population. On the other hand, the three larvae identified as *Anisakis* type II in two hakes originating from the Balearic division, could belong to the species *A. physeteris* according to the results of previous studies carried out in this fishing ground [25,26,38,39,40].

### 3.2. Parasitization in the Viscera and the Flesh

In the three populations of hake analysed, the prevalence of *Anisakis* type I larvae was higher in the viscera than in the flesh. However, in the case of mean abundance, the European Atlantic hake population did not follow the usual pattern, with the larvae burden being higher in the flesh than in the viscera, reaching a total of 31.2 larvae per fish flesh.

The lowest value of parasitization observed in the European Mediterranean hake might be due to the fact that *A. pegreffii* is the most common *Anisakis* type I species in the Balearic division, a species that has been recognized to present a lower capacity of migration to the flesh than other *Anisakis* species, such as *A. simplex* (*s.s.*) [27,40,42,43].

Although almost all parameters of parasitization were higher in the viscera than in the flesh, with the above-mentioned exception, the high indices of parasitization in the edible fish part implies a real risk for consumers who eat undercooked or not conveniently processed fish. Additionally, this risk may increase with the *post-mortem* migration of *Anisakis* larvae from the viscera to the flesh, as will be discussed below.

### 3.3. Influence of the of Fish Catching Season on Anisakis Parasitization

The analysis of the fish catching season on larvae parasitization, mainly on *Anisakis* larvae showed a similar pattern in the three populations of hakes, with the autumn-winter period reaching the highest parasitization parameters, prevalence and mean abundance, compared to the spring-summer period, with or without statistically significant differences. However, these results are not in accordance with those reported in previous studies in this and other fish species previously analysed, which reported mainly, the highest parasitization parameters during the spring season [12,18,22,36,44,45]. Only a few studies reported a higher parasitation in fish caught during the autumn season, such as in the silver hake [1,12].

The reason for seasonal fluctuations in the population dynamics of *Anisakis* spp. Could be associated with the seasonal fluctuations of biotic and abiotic environmental conditions that indirectly influence the migration of aquatic mammals (final hosts), the amounts of parasite eggs shed, and the availability of zooplankton (intermediate *Anisakis* hosts) [30,34,44,46,47,48].

Moreover, these differences are probably related to the seasonal patterns of migration of the fish species [12,48]. Nevertheless, in the opinion of other authors [49], seasonality in prevalence and intensity of *Anisakis* might not be expected as parasite eggs are shed by the final hosts throughout the year and, similarly, parasitation in crustaceans is non-seasonal.

### 3.4. Influence of the Fish Size on Anisakis Parasitization

Fish size, length and weight, influenced the *Anisakis* larvae parasitization parameters of the three hake population analysed, with the only exception being the prevalence in the silver hake. This positive relation between the size (age) of the fish and the increase of parasitization has been reported particularly in the case of hake analysed in various fishing grounds of the Atlantic Ocean and the Mediterranean Sea [12,22,25,26,27,28,30,33,34,36,38,40], although this is not a universal phenomenon in fish, and this correlation might even be negative [22].

This positive correlation between the size of the hake an the increase of *Anisakis* larvae parasitation may be attributed to various factors [2,8,12,22,25,26,30,33,40]: (a) the cumulative parasitization of larvae along time and the higher likelihood of infection; (b)the reinfection over time through the diet of the hake and the fish spent feeding on its prey; (c) the change in the diet of the hake; piscivorous species such as adult hakes are usually more heavily infected with anisakid larvae compared to strict plankton fish feeders, e.g., sardines and anchovies; hake is a top predator that occupies different trophic levels during its life, and its diet shifts from euphausiids (<16 cm), to fishes consumed by larger individuals.

### 3.5. Migration of Anisakis Larvae to the Fish Flesh

In the three populations of hake analysed, the number of days that passed between fishing and the analysis of fish (which corresponds to the theoretical day of consumption) influenced the presence of larvae in the flesh. Other authors have also reported in various populations of hake analysed that *Anisakis* spp. Larvae are able to migrate from the visceral organs to the flesh after the death of the host, suggesting that this movement may be facilitated by the cold storage or processing of uneviscerated fish [12,22,28,31,46,49]. The days between the catch and the sale influenced the *A. simplex* complex presence in the flesh of parasitized fish that mainly feed on euphausiids, such as the blue whiting tend to accumulate anisakid larvae in their body cavity and viscera, although larvae are more abundant in the hypoaxial musculature of piscivorous fish such as hake and cod [18,28,50,51]. Moreover, the accumulation of larvae in the viscera seems to favour their migration to the flesh.

Some authors suggest that factors such as temperature [4,52] or pH [53] play an important role in the post-mortem motility of larvae in the host. However, other experiments suggest that keeping the fish at low temperature could reduce larval migration [43].

Additionally, a negative correlation between the proportion of larvae in the flesh and the fish size was also observed in the three hake populations. Cruz et al. [51] put forward another hypothesis concerning the distribution of *Anisakis* larvae between the flesh and the viscera, proposing that the percentage of larvae in the flesh should be inversely related to the fish size as the migratory distance to the flesh increases with fish size. Intra-vitam migration of *Anisakis* larvae into the flesh is recognizable by the high number of melanized parasitic capsules in the flesh. It seems likely that the proximity of the fish gut and visceral cavity to the antero-ventral musculature facilitates the migration of worms toward this part of the fish but other factors (e.g., related to the immunological response of the fish, condition, density-dependent mechanisms and aggregating behavior) may also contribute to this pattern [22].

Conversely, some authors argue that fish size cannot be used as a reliable predictor of post-mortem larvae migration [1,44], suggesting that larval distribution might be governed by the conditions encountered within host tissues but not by distance.

### 3.6. Assessment of Human Anisakiasis Risk through the Consumption of Hake

The populations of European Atlantic hake and the silver hake can be classified as “high-risk” fish species for human anisakiasis, as previously proposed [12]. Only the European Mediterranean hake, due to its lower parasitization parameters, mainly flesh parasitization, and considering that the sample was caught from the Balearic division, can be classified as intermediate-risk fish species.

Considering the prevalence and abundance registered, all of them could represent a risk of infection when consumed raw, marinated, or poorly cooked, if not being adequately frozen beforehand. Related to this issue, although the disease may appear in different forms, it is necessary to consider the anisakiasis allergy risk due to the allergenic potential of dead larvae, as the presence of thermostable parasite allergens has actually been proved [54].

In general, the proposed human anisakiasis risk through hake consumption is in accordance with other results previously reported, specifically, that carried out by [34] in hake caught in some Northeast Atlantic fishing grounds, classifying the samples from the Grand Sole and Galician coast as FPR Poor Standard (that means very high-risk), and the sample from the Portugal coast as FPR Fair Standard (that means high-risk) of human anisakiasis.

Although anchovies have been recognized as the main fish species causing human anisakiasis, at least in Spain [4], other fish species, such as hake, have also been implicated in human parasitization (e.g.,: 9 cases of 96 gastrointestinal cases, reported in Madrid, after the consumption of undercooked hake [55]; a case of acute anisakiasis, also reported in Madrid, after the ingestion of fried hake and fish ova [56]; and three cases of gastric anisakiasis, reported in Barcelona, caused by the consumption of undercooked hake [24]).

This high-risk of hake has an important impact both on public health and in the fisheries economy, considering that this fish species is the most important demersal fish resource caught off temperate Western Europe, highly priced and mostly sold fresh in European markets [34]. On the other hand, the finding of the other two nematode larvae found parasitizing the three hake populations (i.e., the anisakid *Contracaecum* spp. and the Ascaridoidea *Hysterothylacium*, spp.), although the prevalence observed is relatively low (2–10%), cannot be overlooked, not due to their unclear zoonotic potential with only few human cases reported [57,58,59,60,61] if not mainly due to their deleterious effect at commercial level. Moreover, these two nematode larvae resembling *Anisakis* larvae to a non-expert, pose additional issues associated to the quality of the product. Consequently, consumers should be warned about its presence, especially in the viscera as the consumer’s naked eye is not able to differentiate between these genera and *Anisakis* or other Ascaridoidea nematodes [12,45].

### 3.7. Global Preventive Measures and Recommendations to Consumers to Prevent the Human Anisakiasis Risk

To prevent human anisakiasis, the most effective measure is, according to the recommendations of the Commission Regulation (EU) No1276/2011 [62] and the Spanish Royal Decree 1420/2006 [63], cooking the piece until a core temperature of 60–70 °C for 5–10 min has been reached, or freezing fish at −20 °C at least for 24 h., at −35 °C for at least 15 h, in case of fish meant to be consumed raw or undercooked. Nevertheless, domestic freezers do not always provide a low homogenous temperature and are, therefore, not able to inactivate larvae. The freezing tolerance of these nematodes constitutes a safety risk when domestic freezers operating at low cooling capacity are used for the inactivation of these larvae, as temperatures are not sufficiently low or homogenous [12,45].

Moreover, the increase of the transmission risk through a prolonged time lag between the catch and the consumption of hake suggest the immediate evisceration after capture recommendable as an effective measure to prevent parasite presence [12].

Additionally, consumers must be warned about the presence and the potential effect of the parasite, thus preventing the possible confusion with *Anisakis* spp. larvae [12,45].

## 4. Materials and Methods

### 4.1. Samples and Parasitological Procedures

A total of 536 hake specimens (Table 9), belonging to the species *M. bilinearis* and *M. merluccius*, were purchased in branches of various nationwide Spanish supermarket chains located in the city of València and its metropolitan area. Hake specimens, fresh and uneviscerated, were chosen at random. All available information about the acquired fish specimens was obtained from the fish crates at the selling point. This information consisted mainly of: scientific and common names; date of capture; and FAO (Food and Agriculture Organization of the United Nations) major fishing area, subarea and division. *M. bilinearis* specimens originated from FAO area 21—Atlantic Northwest (n = 172), subareas: 21.2—Labrador coast, 21.3—Newfoundland, and 21.4—Northwest Atlantic. *M. merluccius* specimens caught in the Atlantic originated from FAO area 27—Atlantic Northeast, subarea/division: undefined subarea on fish crate labels (n = 95); subarea 27.7—Iris Sea, West of Ireland, Porcupine Bank, Eastern and Western English Chanel (n = 14), division 27.7.J—Southwest Ireland-East—Grand Sole Bank; subarea 27.8—Bay of Biscay (n = 54), division 27.8.b—Bay of Biscay-Central (n = 47) and division 27.8.c—Bay of Biscay-South (n = 7); subarea 27.9—Portuguese waters (n = 9), division 27.9.a—Portuguese waters-East (n=7), and division 29.9.b—Portuguese waters-West (n = 2). *M. merluccius* caught in the Mediterranean originated from FAO area 37—Mediterranean and Black Sea (n = 192), subarea 37.1—Western Mediterranean, division 37.1.1—Balearic.

All specimens were refrigerated at 4 °C in the laboratory until dissection. Each hake specimen was weighed and the total length was measured. Specimens were classified into two groups according to weight (W1 and W2) and length (L1 and L2). However, different groups of weight and length were established considering that two species of *Merluccius* were analysed and, as above-mentioned, Atlantic and Mediterranean *M. merluccius* populations have to be considered as separated stocks. Moreover, specimens were also classified according to the season of capture in two periods: autumn-winter and spring-summer (Table 9). Additionally, the days (between 1–8 days) passed between the fishing date and the analysis (the hypothetical date of consumption) were also recorded.

As described by [12], the viscera and flesh of each hake were placed in two different Petri dishes and examined separately for the presence of nematode larvae. The viscera were dissected under a stereoscopic microscope and the flesh, after a previous visual inspection, underwent artificial enzymatic digestion [64]. The resulting product from the digestion was also examined under a stereoscopic microscope. Identification of each larva found was based on the morphological characteristics described in the literature [65,66]. The main characteristics considered for these classifications were the position of the excretory pore, the arrangement and separation of the digestive tract into the oesophagus, ventricle and the presence/absence of structures such as intestinal caeca and oesophageal appendix, as well as the shape of the tail. In the case of *Anisakis*, the differentiation between two groups of larvae (*Anisakis* type I and type II) was also carried out. As above-mentioned, molecular diagnosis of each specimen at specific level was not carried out, as the main objective of the study is to inform consumers about the presence of nematode larvae and how to minimize the risk of infection.

### 4.2. Statistical Analysis

The number of parasitized hosts, prevalence (frequency of recording), mean abundance and range of parasitization were analysed according to Bush et al. [67] for the total of nematode Ascaridoidea larvae found, as well as for each of the genera identified. Moreover, these parameters were also calculated for the parasitization site (viscera and flesh) and the season of capture.

The potential risk of human anisakiasis was assessed through the analysis of the influence of intrinsic and extrinsic factors on the prevalence and abundance of larvae identified as *Anisakis* type I. The complete statistical analysis comprised the comparison of prevalence (χ^2^ test) and abundances (Mann-Whitney-U-test, in the case of season of capture, weight and length groups, and Wilcoxon-Z-test, in the case of sites of parasitization). The possible influence of the number of days after capture (which corresponds to the theoretical day of consumption) on the presence of larvae in the flesh was also analysed by Binary Logistic Regression (BLR), considering parasitized hosts only. Furthermore, the Spearman’s rank correlation coefficient (Rho) was also applied to analyse the potential influence of quantitative intrinsic factors (weight and length) on the *Anisakis* type I burden, as well as their influence on the proportion of larvae in the flesh.

Statistical significance was established at *p* < 0.05. Statistical analyses were carried out using the IBM SPSS 26.0 for Windows (International Business Machines Corporation, Armonk, New York, NY, USA) and StatView 5.0 (Statistical Analysis System Institute Inc., Cary, NC, USA) software packages.

## 5. Conclusions

The following recommendations should be considered by consumers to avoid human anisakiasis through the consumption of hake, *Merluccius* spp., in any of its forms: cooking the piece until a core temperature of 60–70 °C for 5–10 min has been reached, or freezing fish at −20 °C at least for 24 h, at −35 °C for at least 15 h, in case of fish meant to be consumed raw or undercooked, and above all of the fish ought to be consumed fresh, avoiding prolonged refrigeration, and it should be eviscerated beforehand, preferably straight after the catch.

Consumers should be informed about the most important variables likely to increase the risk of human anisakiasis (i.e., the traceability data concerning the place and the date of capture), preservation method, as well as the scientific and the common name of the fish, the place of capture (FAO area and division) and the day of capture, available at the fish selling point and included on the crate of the fish, as regulated by the European Commission [68], that should be easily visible for the consumers.

## Figures and Tables

**Table 1 pathogens-11-00622-t001:** Prevalence (P) of Ascaridoidea nematode larvae in the hake (*Merluccius* spp.) sample analysed with respect to its geographic origin.

Hake Species	Total Ascaridoidea		*Anisakis* Type I		*Contracaecum* spp.		*Hysterothylacium* spp.	
	n	P (CI 95%)	n	P (CI 95%)	n	P (CI 95%)	n	P (CI 95%)
Atlantic European hake (*M. merluccius*)	151	88 (83–93)	151	88 (83–93)	17	10 (6–15)	4	2 (1–5)
Mediterranean European hake (*M. merluccius*)	50	26 (17–35)	46	24 (18–31)	4	2 (1–5)	3	2 (1–5)
Silver hake (*M. bilinearis*)	113	66 (57–75)	106	62 (55–69)	5	3 (1–7)	18	10 (6–15)

n, number of parasitized fish specimens; CI, confidence interval.

**Table 2 pathogens-11-00622-t002:** Mean abundance (mA) and range of Ascaridoidea nematode larvae in the hake (*Merluccius* spp.) sample analysed with respect to its geographic origin.

Hake Species	Total Ascaridoidea	*Anisakis* Type I	*Contracaecum* spp.	*Hysterothylacium* spp.
	mA (SE) (Range)	mA (SE) (Range)	mA (SE) (Range)	mA (SE) (Range)
Atlantic European hake (*M. merluccius*)	51.7 (14.660) (1–1951)	51.3 (14.635) (1–1950)	0.4 (0.134) (1–18)	0.1 (0.105) (1–18)
Mediterranean European hake (*M. merluccius*)	4.7 (1.350) (1–182)	4.6 (1.351) (1–182)	0.03 (0.014) (1–2)	0.03 (0.016) (1–2)
Silver hake (*M. bilinearis*)	1.9 (0.186) (1–12)	1.7 (0.177) (1–12)	0.03 (0.013) (1)	0.2 (0.038) (1–3)

SE, standard error.

**Table 3 pathogens-11-00622-t003:** Prevalence (P) of *Anisakis* type I in the hake (*Merluccius* spp.) sample analysed with respect to its geographic origin, microhabitat, and its season of capture.

Hake Species	Viscera		Flesh		Autumn-Winter		Spring-Summer	
	n	P (CI 95%)	n	P (CI 95%)	n	P (CI 95%)	n	P (CI 95%)
Atlantic European hake (*M. merluccius*)	131	76 (70–82)	113	66 (59–73)	63	93 (87–99)	88	85 (78–92)
Mediterranean European hake (*M. merluccius*)	40	21 (16–27)	30	16 (11–22)	30	27 (21–34)	16	20 (15–26)
Silver hake (*M. bilinearis*)	88	51 (44–59)	46	27 (21–34)	73	65 (53–77)	33	55 (46–64)

n, number of parasitized fish specimens; CI, confidence interval.

**Table 4 pathogens-11-00622-t004:** Mean abundance (mA) and range of *Anisakis* type I in the hake (*Merluccius* spp.) sample analysed with respect to its geographic origin, microhabitat, and its season of capture.

Hake Species	Viscera	Flesh	Autumn-Winter	Spring-Summer
	mA (SE) (Range)	mA (SE) (Range)	mA (SE) (Range)	mA (SE) (Range)
Atlantic European hake (*M. merluccius*)	20.1 (5.449) (1–830)	31.2 (9.939) (1–1120)	92.4 (34.625) (1–1950)	24.4 (7.771) (1–574)
Mediterranean European hake (*M. merluccius*)	3.6 (1.142) (1–167)	1.1 (0.289) (1–39)	6.2 (2.159) (1–182)	2.5 (1.250) (1–90)
Silver hake (*M. bilinearis*)	1.3 (0.162) (1–12)	0.4 (0.059) (1–6)	2.0 (0.244) (1–12)	1.1 (0.206) (1–9)

SE, standard error.

**Table 5 pathogens-11-00622-t005:** Statistically significant differences found between weight groups and the prevalence and mean abundance of *Anisakis* type I of the three populations of hake (*Merluccius* spp.) analysed.

Hake Species	χ^2^	*p*	OR (95% CI)	U	*p*
Atlantic European hake (*M. merluccius*)	15.799	<0.0001	16.39 (2.14–125.24)	735.5	<0.0001
Mediterranean European hake (*M. merluccius*)	12.174	<0.0001	3.38 (1.70–6.73)	2917.5	<0.0001
Silver hake (*M. bilinearis*)	---	---	---	2864.5	0.010

χ^2^, Chi square test value; *p*, *p* value; OR, odds ratio; CI, confidence interval; U, Mann-Whitney test value.

**Table 6 pathogens-11-00622-t006:** Statistically significant differences found between length groups and the prevalence and mean abundance of *Anisakis* type I of the three populations of hake (*Merluccius* spp.) analysed.

Hake Species	χ^2^	*p*	OR (95% CI)	U	*p*
Atlantic European hake (*M. merluccius*)	10.108	0.0015	9.13 (2.05–40.57)	746.5	<0.0001
Mediterranean European hake (*M. merluccius*)	4.363	0.0037	2.04 (1.04–3.99)	3804.5	0.008
Silver hake (*M. bilinearis*)	---	---	---	2929.5	0.032

χ^2^, Chi square test value; *p*, *p* value; OR, odds ratio; CI, confidence interval; U, Mann-Whitney test value.

**Table 7 pathogens-11-00622-t007:** Statistically significant correlations found between quantitative values of weight and length and the mean abundance of *Anisakis* type I of the three populations of hake (*Merluccius* spp.) analysed.

Hake Species	Weight		Length	
	Rho	*p*	Rho	*p*
Atlantic European hake (*M. merluccius*)	0.679	<0.0001	0.695	<0.0001
Mediterranean European hake (*M. merluccius*)	0.324	<0.0001	0.307	<0.0001
Silver hake (*M. bilinearis*)	0.311	<0.0001	0.318	<0.0001

Rho, Spearman’s rank correlation coefficient value; *p*, *p* value.

**Table 8 pathogens-11-00622-t008:** Statistically significant correlations found between quantitative values of weight and length and the proportion of *Anisakis* type I larvae present in the flesh of the three populations of hake (*Merluccius* spp.) analysed.

Hake Species	Weight		Length	
	Rho	*p*	Rho	*p*
Atlantic European hake (*M. merluccius*)	−0.249	0.008	−0.227	0.015
Mediterranean European hake (*M. merluccius*)	---	---	−0.380	0.0038
Silver hake (*M. bilinearis*)	−0.406	0.005	−0.411	0.005

Rho, Spearman’s rank correlation coefficient value; *p*, *p* value.

**Table 9 pathogens-11-00622-t009:** Description of the hake (*Merluccius* spp.) sample analysed by season of capture, weight range, weight groups, length range, and length groups.

Hake Species	Total (n)	Autumn-Winter	Spring-Summer	Weight Range (g)	Weight Group	n	Length Range (cm)	Length Group	n
Atlantic European hake (*M. merluccius*)	172	68	104	69.7–549.6	W_1_ ≤ 200 g	103	21.7–41.5	L_1_ ≤ 30 cm	96
				(x = 198.7)	W_2_ > 200 g	69	(x = 30.3)	L_2_ > 30 cm	76
Mediterranean European hake (*M. merluccius*)	192	110	82	37.0–582.0	W_1_ ≤ 150 g	129	18.4–41.1	L_1_ ≤ 27 cm	105
				(x = 151.4)	W_2_ > 150 g	63	(x = 27.2)	L_2_ > 27 cm	87
Silver hake(*M. bilinearis*)	172	60	112	51.4–215.9	W_1_ ≤ 100 g	93	19.0–32.5	L_1_ ≤ 25 cm	72
				(x = 100.8)	W_2_ > 100 g	79	(x = 25.7)	L_2_ > 25 cm	100

## Data Availability

The data presented in this study are available on request from the corresponding author.

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
