# Peer review of "Anisakid Nematodes and Potential Risk of Human Anisakiasis through the Consumption of Hake, Merluccius spp., Sold Fresh in Spanish Supermarkets"

_pathogens, 2022, doi:10.3390/pathogens11060622_

Round 1
Reviewer 1 Report
The manuscript entitled „Anisakid nematodes and potential risk of human anisakiasis through the consumption of hake, Merluccius spp., sold fresh in Spanish supermarkets” presented to me for review is important, it proves the need for constant monitoring of the anisakids-infestation of economically important fish in order to reduce the risk of human anisakidosis. The manuscript is well written, I recommend it for printing after minor corrections.
Critical remarks
While there is an obligation to include information about fish (catching place etc.) on crate labels, I am afraid that they are not important to most consumers as they do not understand what this information means in practice and why it is important to them. It would be more important that the institutions dealing with trade in fish and safety of food admitted to the market were informed about the parameters of infestation of fish from given fishing areas - this should be added in the manuscript as the third goal of the work
Minor corrections
Line 100:
… potential source of human anisakiosis, with some human cases having been reported due to its consumption, such as the three recently reported cases in Barcelona [24].
Note: I understand the use of the term "anisakiosis" (after the author of the cited publication) but in your own text you should use one of these terms throughout the manuscript.
Line 243:
„2.3. Global parasitization ”
Note: it should probably be subsection 3.1, not 2.3
Lines: 250-252
„The very high global prevalence of both ascaridoid and Anisakis type I larvae in the sample of European Atlantic hake, the high prevalence of the sample of silver hake, and the low prevalence of the sample of European Mediterranean hake stands out.”
Note: incomprehensible sentence
Lines 434-443
Notes:
- inconsistency in nomenclature (FAO area 21, while FAO zone 27 and FAO zone 37),
- inconsistency in the reporting of the number of fish from individual subareas (if information was not always available, please write it down)
Reviewer 2 Report
Interesting results from an epidemiological point of view.
The paper has to be revised regarding statistic presentation. The data are representative and results are understandable and are ideal to perform a multidimensional analysis. However the authors prefer a classic parametric analysis. No objection is raised by this reviewer regarding that. In results and discussion sections the data and conclusions after statistics performance are reflected.
I would suggest to use “Ascaridoidea” intead “Ascaridoid”. Families Anisakidae (Anisakis and Contracaecum genera in this case) and Rhaphidascaridae are members of Superfamiy Ascaridoidea. There are some misunderstandings in the text, regarding that, as for instance line 250 (“both ascaridoid and Anisakis type..”); Anisakis is member of Ascaridoidea: clarify.
In Material and Methods, clarify that prevalence refers to frequency of recording.
I would like to do some consideration regarding tables1 and Tables 5, 6 and 9:
Table 1: Total Ascaridoidea is the addition of Anisakis Type I plus Contracaecum spp and Hysterothylacium spp…..172 (Atlantic European hake); 53 (Mediterranean European hake); 129 (Silver hake). That is important in order to calculate the Odds Ratio (positive or negative association) and X2.
Tables 5, 6 and 9: These tables have to be redo. Tables 5 and 6 deal apparently on quantitative parameters (wight and length; line 196): However it also deal on qualitative one (prevalence): this paragraph (line 196 to 201) has to be rewritten and refer also to data of Table 9 and do not leave to free interpretation of the reader).
I would suggest to combine table 9 with Table 5 and Table 6 forming a new Table 5 where statistical parameters of current Tables 5 and 6 be included. To do that, add five new lines to current Table 9 [these lines being X2, P and OR (95%CI) for qualitative parameters and U and P for quantitative parameters].
Former Tablees 7 and 8 would be new Table 6 and Table 7
